# Phenolic Compounds of Rose Hips of Some *Rosa* Species and Their Hybrids Native Grown in the South-West of Slovenia during a Two-Year Period (2020–2021)

**DOI:** 10.3390/foods12101952

**Published:** 2023-05-11

**Authors:** Nina Kunc, Metka Hudina, Gregor Osterc, Jože Bavcon, Blanka Ravnjak, Maja Mikulič-Petkovšek

**Affiliations:** 1Department of Agronomy, Biotechnical Faculty, University of Ljubljana, Jamnikarjeva 101, 1000 Ljubljana, Slovenia; 2University Botanic Garden, Biotechnical Faculty, University of Ljubljana, Ižanska cesta 15, 1000 Ljubljana, Slovenia

**Keywords:** rosa, *Rosa corymbifera*, *Rosa glauca*, *Rosa subcanina*, *Rosa gallica*, phenolic compounds, HPLC, extraction

## Abstract

The genus *Rosa* is very extensive and variable, so it remains very unpredictable and uninvestigated. This also holds true for values of secondary metabolites in rose hips, which are important for several purposes (human diet, protection of plants against pests, etc.). The aim of our study was to determine the content of phenolic compounds in the hips of *R.* × *R. glauca*, *R. corymbifera*, *R. gallica* and *R. subcanina*, which grow wild in nature in southwestern Slovenia. We examined the content of phenolic compounds in different parts of rose hips, in the flesh with skin and in the seeds, depending on the individual species, over a period of two years, 2020 and 2021. We also considered the influence of environmental conditions on the content of the mentioned compounds. In both years, the content of phenolic compounds was higher in the flesh with skin than in the seeds. Considering the total content of phenolic compounds in the flesh with skin, *R. gallica* stands out (15,767.21 mg/kg FW), but the hips of this species accumulated the lowest number of different phenolic compounds. The lowest content of total phenolic compounds (TPC) was found in *R. corymbifera,* in the year 2021 (3501.38 mg/kg FW). The content of TPC (in both observed years) in the seeds varied between 1263.08 mg/kg FW (*R. subcanina*) and 3247.89 mg/kg FW (*R.* × *R. glauca*). Among the anthocyanins, cyanidin-3-glucoside was determined, which was predominant in *R. gallica* (28.78 mg/kg FW), and at least was determined in *R. subcanina* (1.13 mg/kg FW). When comparing the two years of the period (2020–2021), we found that 2021 was more favorable for the formation of phenolic compounds in the seeds, but 2020 in the flesh with skin.

## 1. Introduction

The *Rosa* genus is very extensive and numbers about 200 species. Only 26 of these species are native to Slovenia, but their identification and recognition are not easy. The intraspecies and interspecies diversity has been characterized by many of them. Their diversity is influenced by their occurrence in all phytogeographical regions of Slovenia. *Rosa corymbifera* grows in nature in the northern, eastern, and southwestern parts of Slovenia, while *Rosa glauca* is distributed throughout Slovenia, except on the northwestern side of the country. *R. corymbifera*, *R. glauca*, *R. gallica* and *R. subcanina* thrive in Slovenia only in the northern and southeastern parts of the country [1,2].

Interest in the commercial development of plants as sources of antioxidants has recently increased worldwide. They are increasingly used as additives to improve the properties of foods for nutritional and preservation purposes and to prevent oxidation-related diseases. These plants include roses [3], which are considered an excellent source of polyphenolics and vitamin C [4]. Their antioxidant activity is due to their ability to chelate metals, inhibit lipoxygenase and scavenge free radicals [5]. They prevent the negative reactivity of the resulting unwanted reactive oxygen/nitrogen species with metabolic activities in the body. These compounds are used to treat hypertension, neurodegenerative diseases, inflammatory infections, arthritis, renal diseases, skin diseases and metabolic disorders. Inhibition of the enzyme that hydrolyzes carbohydrates is used to treat type 2 diabetes. Inhibition of cholinesterase is used to treat Alzheimer’s disease [6,7,8,9]. Extracts of *R*. *canina* have shown antiarthritic, anti-inflammatory, analgesic, antidiabetic, autoprotective, antimicrobial and immunomodulatory properties [9,10].

Phenolic substances, together with other chemicals, are involved in most of the listed effects. The main phenolic compounds found in rose hips are tannins, flavonoids, phenolic acids and anthocyanins. The literature provides various quantitative and qualitative descriptions of the phenolic profile of roses. Much is known about *R. canina*, but very little is known about the phenolic profiles of *R. glauca*, *R. corymbifera*, *R. gallica* and *R. subcanina* [11]. Therefore, we will also list the values of other roses. Koczka et al. [11] determined the content of phenolic compounds in four species of *Rosa*. They were *R*. *canina*, *R*. *gallica*, *R*. *rugosa* and *R*. *spinosissima*, which were collected from two places in Hungary. They found that of all the rose hips studied, *R*. *spinosissima* had the highest phenolic content. Ethanol and water extracts were included in the study, and it was found that ethanol extracts were more suitable and effective for the extraction of the phenolic compounds of the *Rosa* genus. Fascella et al. [12] studied four selected Sicilian rose hips: *R*. *canina*, *R*. *corymbifera*, *R*. *micrantha* and *R*. *sempervirens*. In their studies, they found that the selected rose hips showed a strong variability. The highest content of phenolic compounds was found in *R*. *canina* and *R*. *sempervirens*. Galloyl derivatives, catechin derivatives, ellagic acid and quercitrin were mentioned as the main phenolic compounds. They confirmed that the aforementioned rose hips are a rich source of phenolic compounds, making them a potential functional food. The phenolic composition was also reported by Ersoy et al. [13], who included *R*. *canina* from Turkey in their studies. They confirmed that phenolic compounds are one of the most important compounds in rose hips, which have an antioxidant role in rose hips fruits. Phenolic compounds were considered to be the main group of compounds contributing to the antioxidant activity of the fruit. Using the HPLC method, Butkevičiūtė et al. [14] quantitatively and qualitatively analyzed the phenolic composition of different rose hips of *Rosa* L. They found that the dominant compound was catechin, which was predominant in *R*. *subcanina*. The highest total phenolic compound content was determined in samples of *R*. *pisocarpa*. Tumbas et al. [3] and Hosni et al. [15] determined quercetin and ellagic acid as the major phenolics of *R*. *canina*, while Tuerkben et al. [16] and Olsson et al. [17] reported that the major phenolic components in species without ellagic acid or kaempferol were quercetin and catechin. Sun et al. [18] identified 531 phenolic compounds in the fruits of *R*. *xanthina* f. spontaneously during fruit ripening using UPLC-MS/MS. Phenolic acids and flavonoids dominate as the main part of the total composition of phenolic compounds. They reported that the total polyphenolic content of rose hips was higher than that of some other fruits and berries, such as blueberries, strawberries, and raspberries. Cunja et al. [19] determined by HPLC/MS, the secondary metabolites of *R. canina*, harvested at different stages of ripeness. They identified 45 different phenolic compounds. They accounted for 90% of the total flavanols and proanthocyanins, which did not change significantly with ripening. However, they found that the content of catechins, phloridzin, flavanones, and quercetin glycosides was highest in the early stages of harvest and decreased after frost. However, the total phenolic content increased until the middle of the harvest and then began to decrease.

There have been some additional studies related to the phenolic profile of *R. canina,* but extremely little is known about the content and composition of the phenolic compounds of the hips of various other main rose species. Various phenolic compounds are known to be substances with antioxidative potential and also as substances with other beneficial effects on human health (some reports have even connected high phenolic amounts with lower allergic potential). The composition of the hips of these main species as a source of phenolics has therefore increased in recent years due to the search for healthy food of autochthonous origin. Additionally, many of these main species are parents of various new rose cultivars, and some of them are also cultivated for food processing purposes. If we want to know the characteristics of these cultivars, it is very important to know the features of the parents. The main purpose of our research was to determine the content of phenolic substances in the fruits of different main species of roses widespread on the territory of Slovenia. Included in the experiment were *R. corymbifera*, *R. gallica*, *R. subcanina* and an unknown cross with *R. glauca,* all of which grow naturally. Detailed analysis of the rose hips was performed with respect to various phenolic compounds. The phenolic compounds were measured separately in the pulp and seed of the rose hips.

## 2. Materials and Methods

### 2.1. Plant Material

The study included rose hips from *R. corymbifera*, *R. subcanina* and *R. gallica* and an unknown cross with *R. glauca*, which were collected in the southwestern part of Slovenia, more precisely in the vicinity of Nanos, Zagorje, near Pivka and Podgorje (Figure 1 and Figure 2).

*R. corymbifera* is a 1 to 3 m tall shrub that blooms from May to June, and the fruits ripen from October to November. *R. glauca* is also 1 to 3 m tall. It blooms from June to July and ripens in early September. *Rosa gallica* is found in light forests, abandoned meadows, pastures, shrubs and along pathways in many regions in Slovenia. It is a 0.3 to 1 m tall, branched shrub that flowers in June and July, and the fruits ripen in September. *R. subcanina* was used as a control species. *R. subcanina* is visually very similar to *R. canina* (which is often taken as a control species), except that the leaflets are at least partially hairy along the veins and stalk, and it is much more widespread in the region of our study than *R. canina*. *R. subcanina* is the tallest of all the studied species, between 1.5 and 2 m. It flowers from May to July, and the fruits ripen from September to October [1]. All hips were harvested in the years 2020 and 2021 at full ripeness, stage BBCH 88, according to Meier et al. [20]. All plants in each area grew under similar climatic conditions, in a narrow area, on the sunny side of slopes. The collected hips were placed on ice and immediately brought to the laboratory, where the material was stored at −20 °C until further analyses.

### 2.2. Agrometeorological and Pedological Data

The pedological conditions of the soil were described with the help of a pedological map of Slovenia and other relevant literature. Using the archive of measurements, we examined the agrometeorological conditions in the mentioned period. The solar radiation was also measured at the location itself using an SRI-2000-UV Spectrophotometer.

Zagorje near Pivka belongs to the Municipality of Postojna, and Nanos to the Municipality of Vipava. It is a rugged area located in northern Karst, in the SW part of Slovenia, and it covers 270 km^2^. More than 60% of the area is covered by forest. Nanos is a high karst plateau that separates the inland part of Slovenia from Primorje. On the eastern side, it is separated from the karst plateau Hrušica by a foreland fault, and on the west, it separates the Bela valley from the forest plateau of Trnovo. Steep slopes, 500 to 700 m high, on the western side, separate it from the Vipava and Pivka basins. Altitudes around 1100 m predominate, with the western half being lower (800–950 m), with less rock. The alluvium consists of Cretaceous and Jurassic limestones, which are cut into the flysch rocks of the Vipava valley as a horizontal fold, on which basic brown carbonate soils and rendzinas have developed. Due to the scrub, the weathering thickness is very uneven and heavily exposed to washing. In Zagorje near Pivka, a karst aquifer with a limestone base predominates, which allows water to flow underground through wide cracks and channels in different directions towards springs on the outskirts. This area is characterized by a fluvial-denudation relief [21,22].

These are areas of the hinterland with a moderate Mediterranean climate and a moderate continental climate belonging to the Submediterranean and Dinaric phytogeographic regions in western and southern Slovenia. In Zagorje near Pivka (Figure 3), the lowest measured temperature was in January 2021 (0.9 °C) and the highest in July 2021 (20.5 °C). The winter of 2020 was colder than the winter of 2021, and the summer in the second part was also slightly warmer. The stormiest days were recorded in August 2020, 10 days. Ten days was also the highest number of days on which the area was covered with snow, which was in January 2021. In year 2021, we observed different distributions of precipitation than in year 2020. The highest amount of precipitation was recorded in May 2021, 300.8 mm. The driest month was June 2021, with only 14.7 mm of precipitation [23].

On Nanos (Figure 4), the temperatures in the winter months were lower in 2021. The lowest temperatures were recorded in January 2021, −4 °C. Summer temperatures ranged between 12 and 15.8 °C in 2020 and between 13.4 and 15 °C in 2021. In the summer months of 2020, a total of 8 stormy days were recorded, which was also the only stormy period during our observation. In 2021, as many as 70 days had snow cover, and in 2020 only 13. The largest amount of precipitation was in May 2021, 379.6 mm. The driest month was June 2021, with 33.7 mm of precipitation [23].

In Podgorje (Figure 5), the only month with temperatures below 0 °C was January 2021. The highest temperatures were recorded in August 2020 and June 2021; in both months, it was 18.5 °C. Storms were mostly distributed throughout the summer months. Only three days in December 2021 had snow cover. It was noticeable that there was little precipitation during the year. The most was 260 mm in December 2020, when they also recorded the only snowy days during our observation. The lowest amount of precipitation was in January 2020, only 10.4 mm [23].

The total photosynthetic active radiation (PAR) was the highest in Nanos (1655.6 W/m^2^) and the lowest in Podgorje (1481 W/m^2^) (Table 1). The light intensity in the range of 400–500 nm was, on average, the highest in the Nanos area (544.7 W/m^2^) and the lowest in Podgorje (423.9 W/m^2^). Average light intensity in the range of 500–600 nm was between 535.1 W/m^2^ (Podgorje) and 559.4 W/m^2^ (Zagorje near Pivka). In the range of 600–700 nm, the highest light intensity on average was detected in Zagorje near Pivka, 603.2 W/m^2^.

### 2.3. Extraction and Analysis of Phenolic Compounds

The rosehip extraction method (separated flesh with skin and seeds) was performed according to the instructions previously described by Kunc et al. [24]. Analyzes were performed in triplicate. The extraction took place in such a way that each sample was individually ground into powder with the addition of liquid nitrogen with the help of a mortar. A determined mass (between 0.2 g and 1.3 g, depending on the amount of material available) of the sample was then weighed into a centrifuge, to which the extraction solution was then added. The extraction solution was 3%. The centrifuges were then placed in a cooled ultrasonic bath (Iskra PIO, SONIS 4 GT, Šentjernej, Slovenia), where extraction took place on ice for 1 h. The samples were then centrifuged with an Eppendorf 5810 R centrifuge for 7 min at 10,000× *g* and 4 °C. After centrifugation, the supernatants were filtered through a 0.20 mm polyamide/nylon filter (Macherey–Nagel, Düren, Germany) into labeled vials. Vials were stored at −20 °C until further analysis of phenolic compounds on HPLC.

A Thermo Scientific Dionex HPLC system with a diode array detector (Thermo Scientific, San Jose, CA, USA) interfaced with Chromeleon workstation software was used for the analysis of phenolic compounds. The chromatographic method previously described by Mikulic-Petkovsek et al. [25] was used for the analysis. The wavelengths of the detector were 280 nm, 350 nm and 530 nm. The mobile phases used in procedures were A: 3% acetonitrile/0.1% formic acid/96.9% bi-distilled water; B: 3% water/0.1% formic acid/96.9% acetonitrile. Gradient elution of both mobile phases was carried out according to the previous description of Mikulic-Petkovsek et al. [26]. The flow rate of the mobile phases was 0.6 mL/min. The column used for the procedure itself was a Gemini C18 (150 × 4.6 mm 3 μm; Phenomenex, Torrance, CA, USA) heated to 25 °C.

Phenolic compounds were identified using electrospray ionization (ESI) mass spectrometer (LTQ XL Linear Ion Trap Mass Spectrometer, Thermo Fisher Scientific, USA). The mentioned spectrometer works in positive (anthocyanins) or negative (all other phenolics) ionization mode. All mass spectrometer conditions were set according to the instructions of Mikulic-Petkovsek et al. [26]. Spectral data were processed using Excalibur software (Thermo Scientific). The identification of compounds was confirmed by comparison of retention times and their spectra, by the addition of a standard solution to the sample, and by fragmentation and comparison with literature data. For the quantification of phenolics in rosehips we used followed standards: procyanidin B1, *p*-coumaric acid, catechin, epicatechin, gallic acid, caffeic acid, ferulic acid, sinapic acid, naringenin, quercetin-3-rutinoside, quercetin-3-galactoside, quercetin-3-glucoside, quercetin-3-xyloside, cyanidin-3-glucoside, apigenin-7-glucoside, kaempferol-3-glucoside, phloridzin, quercetin-3-arabinopyranoside, quercetin-3-arabinofuranoside, isorhamnetin-glucoside, chlorogenic acid, phloretin, quercetin-3-rhamnoside, myricetin-3-rhamnoside and 4-caffeoylquinic acid. For phenolic compounds of which the standards were not available, contents were presented as equivalents of chemically similar phenolics (all quercetin derivatives as quercetin-3-glucoside, both sinapic acid hexosides as sinapic acid, 3-feruloylquinic acid as ferulic acid, naringenin hexosides as naringenin, all isorhamnetin derivatives as isorhamnetin-3-glucoside, taxifolin pentosides as quercetin-3-galactoside, all kaempferol derivatives as kaempferol-3-glucoside, both apigenin derivatives as apigenin-7-glucoside, both phloretin derivatives as phloretin, all procyanidin derivatives as procyanidin B1, all *p*-coumaroyl derivatives as *p*-coumaric acid, all HHDP derivatives, galloyl- and gallate glycosides as gallic acid. The contents of phenolic compounds were obtained by calculation from the peak areas of the samples and their corresponding standards [24] and expressed in mg per kg fresh weight (mg/kg FW).

### 2.4. Statistical Analysis

Data were compiled using Microsoft Excel 2016 and R commander and expressed as mean ± standard error (SE). One-way analysis of variance (ANOVA) was used to determine significant differences between genotypes within a year. Tukey tests were used to determine significant differences between treatments, and statistical means were calculated at a 95% confidence level (α < 0.05) to determine the significance of the differences. Two-way variance analysis (ANOVA) was used for the determination of statistical differences between genotype and year. Each flesh with skin and seeds was analyzed separately. The shrubs of *R. subcanina* and *R. gallica* did not produce hips in 2020. Unfortunately, it was impossible to collect data on the content of the analyzed compounds for the two genotypes this year.

## 3. Results

### Phenolic Compounds

The total content of phenolic compounds (Table 2 and Figure 6) in the skin with flesh ranged between 3501.38 mg/kg FW (*R. corymbifera*) and 15,767.21 mg/kg FW (*R. gallica*) in 2021. In seeds, the values were on average lower, ranging from 1263.08 mg/kg FW (*R. subcanina*) to 3247.89 mg/kg FW (*R.* × *R. glauca*). In 2020, the total content of phenolic compounds in the skin with flesh was higher in *R. corymbifera* (12,832.94 mg/kg FW) than the content in the unknown cross with *R. glauca* (7650.93 mg/kg FW). The content in the seeds of the cross with *R. glauca*, however, was higher (1821.24 mg/kg FW) in the mentioned year compared to the total content of phenolics in the seeds of *R. corymbifera* (1285.14 mg/kg FW).

When comparing the years 2021 and 2020, the total content of phenolic compounds in the flesh with skin in both genotypes was significantly higher in 2020. In the seeds, it was the other way around; the value in both genotypes was higher in 2021 (Table 2).

We determined 10 compounds that are classified as hydroxybenzoic acid derivatives (HBA) (Appendix A). Among these, the most compounds were detected in the seeds of *R.* × *R. glauca* and the least, only two, in the seeds of *R. gallica*. In HBA, significant differences between years were observed in the flesh with skin in hips of *R.* × *R. glauca* and in the seeds of hips in *R. corymbifera*. In both cases, these values were higher in 2020. In the skin with flesh from *R.* × *R. glauca*, the content was 70.30 mg/kg FW in 2021 and 222.05 mg/kg FW in 2020. In the seeds of *R. corymbifera*, the content was 43.65 mg/kg FW in 2021 and 62.50 mg/kg FW in 2020. Taxifolin pentosides 1–3 were identified only in the flesh with a skin of *R. corymbifera* and *R. subcanina*. Ellagic acid rhamnoside (*R. subcanina*) and methyl ellagic acid pentoside 1 (*R.* × *R. glauca* and *R. corymbifera*), and ellagic acid pentoside 2 (*R.* × *R. glauca*) were detected only in the seeds. Galloyl quinic acid was present only in *R.* × *R. glauca*, both in the flesh with skin and in the seeds.

Additionally, 13 compounds classified as hydroxycinnamic acid derivatives (HCA) were identified (Appendix A). Among these, most were present in the flesh with a skin of *R.* × *R. glauca* hips and some in the seeds of *R. gallica* hips. In all analyzed samples, the HCA content was higher in the flesh with skin than in the seeds, except in the case of *R. subcanina,* in which the values of compounds from this group were slightly higher in the seeds (328.69 mg/kg FW) than in the flesh with skin (202.30 mg/kg FW). The smallest number of identified HCAs was found in the seeds of *R. gallica* hips, with only five compounds. The largest number of the mentioned compounds was identified in the flesh with a skin of the hips of *R*. × *R. glauca*, 11 compounds. The compounds that appeared only in the flesh were sinapic acid hexoside 2 and *p*-coumaric acid hexoside 2. Sinapic acid hexoside 3 was identified only in the seeds and flesh with a skin of *R.* × *R. glauca* hips. Only the hips of *R.* × *R. glauca* contained 3 feruloylquinic acid in the flesh with skin and in the seeds. Trigalloyl quinic acid 1 was present only in the seeds of *R.* × *R. glauca* and *R. subcanina* hips, whereas trigalloyl quinic acid 2 was present only in the seeds of *R. subcanina* hips. There was no great difference in the HCA content between individual years. The greatest difference was observed in *R. corymbifera*, in which the total HCA content in the flesh with skin was 171.79 mg/kg FW in 2021 and 769.16 mg/kg FW in 2020.

The total content of gallotannins (Appendix A) in the flesh with skin in 2021 was the lowest in *R. corymbifera* (261.61 mg/kg FW) and the highest in *R. subcanina* (912.96 mg/kg FW). The values were lower in seeds, varying between 141.40 mg/kg FW (*R. corymbifera*) and 282.94 mg/kg FW (*R. subcanina*). As can be seen, in 2021, *R. subcanina* had the highest content of the above phenolic compounds, both in the flesh with skin and in the seeds. Comparing the years 2021 and 2020, we observed that the content of gallotannins in the flesh with a skin of *R.* × *R. glauca* was higher in 2020 (681.30 mg/kg FW) than in the following year (377.36 mg/kg FW). The content in the seeds did not change, amounting to 270 mg/kg FW in both years. The content of gallotannins in *R. corymbifera* was also higher in 2020. At that time, it was 1396.09 mg/kg FW in the flesh with the skin, but in 2021, it dropped to only 261.61 mg/kg FW. In seeds, however, the situation was the opposite. In 2021, the content of the mentioned phenolic compounds was 141.40 mg/kg FW, and in 2020, it was only 45.70 mg/kg FW. The highest number of gallotannins (9) was identified in the seeds of *R.* × *R. glauca*. A compound that appeared only in these seeds was digalloyl quinic acid 2. In addition, trigalloyl hexosides 1–3 and methyl gallate acetyl dihexoside, which appeared only in the seeds of *R. corymbifera*, were also present only in the seeds. Digalloyl pentoside was only detected in the flesh with skin in *R.* × *R. glauca* hips.

We identified 10 ellagitannins in hips of the genotypes studied (Appendix A). The number of individual-specific ellagitannins did not differ significantly among species; only *R. corymbifera* stood out. We could detect only four ellagitannins in its seeds. The major difference between the observed years was found for the flesh with a skin of *R. corymbifera*, with which the value of ellagitannins in 2020 was 911.55 mg/kg FW, while in 2021, it was more than half lower, with only 366.51 mg/kg FW. The lowest content of this compound in seeds was detected in *R. corymbifera*, 151.34 mg/kg FW in 2020 and 213.50 mg/kg FW in 2021. Among compounds classified as ellagitannins, only HHDP digalloyl hexoside isomer 1 was detected in the seeds of *R. subcanina* and HHDP galloyl hexoside 2 in the seeds of *R.* × *R. glauca* and in the seeds of *R. subcanina*.

The content of flavanols (Appendix A) was generally higher in the flesh with skin than in the seeds. In 2021, the content in the flesh with skin was highest in the hips of *R. gallica*, 12,293.14 mg/kg FW, and lowest in the hips of *R. corymbifera* (2579.61 mg/kg FW). In 2020, the flavanol content was higher in hips of *R. corymbifera* (9358.54 mg/kg FW) compared to the values in hips of *R.* × *R. glauca* (5848.32 mg/kg FW). In seeds, the values in 2021 ranged between 295.23 mg/kg FW (*R. gallica*) and 2145.76 mg/kg FW (*R.* × *R. glauca)*. In 2020, the content of flavanols in the seeds of *R.* × *R. glauca* (1119.43 mg/kg FW) was higher than in the seeds of *R. corymbifera* (795.11 mg/kg FW).

Comparing the content of the predominant procyanidin dimer 1 between the analyzed years (Appendix A), it can be seen that its content, both in the seeds and in the flesh with skin, was slightly higher in 2021. The catechin content was higher in *R.* × *R. glauca* hips in 2020 (612.31 mg/kg FW) than in 2021 (241.98 mg/kg FW). In the case of *R. corymbifera*, in which catechin was the predominant component, the content was quite similar between years, 116.03 mg/kg FW in 2021 and 191.62 mg/kg FW in 2020.

The content of total flavonols in the flesh with skin ranged between 55.69 mg/kg FW (*R. subcanina*) and 157.94 mg/kg FW (*R. gallica*) in 2021. In 2020, however, these values were between 76.87 mg/kg FW (*R.* × *R. glauca*) and 273.28 mg/kg FW (*R. corymbifera*). Flavonols values in seeds ranged between 12.63 mg/kg FW (*R. subcanina*) and 526.86 mg/kg FW (*R. corymbifera*) in 2021, while in 2020, the values were quite similar and hovered around 90 mg/kg FW on average (Appendix A).

Among flavones, apigenin derivatives 1 and 2 were found, which were present in the flesh with skin. Only apigenin derivative 1 was present in the seeds of *R.* × *R. glauca* hips and in the flesh with a skin of *R. gallica*. We did not detect the mentioned flavones in the seeds of *R. corymbifera, R. subcanina* or *R. gallica*. The values in the flesh with skin were quite similar in the two analyzed years, ranging between 0.12 mg/kg FW (*R. gallica*) and 8.77 mg/kg FW (*R. subcanina*). In the seeds of *R.* × *R. glauca*, the value of apigenin derivative 1 reached 5.04 mg/kg FW in 2021 and 1.16 mg/kg FW in 2020 (Appendix A).

From the group of dihydrochalcones, we determined phloridzin, which was present in the flesh with a skin of *R.* × *R. glauca*, *R. corymbifera*, the seeds of *R. subcanina* and in the seeds of *R. gallica*, in which a high value (1098.45 mg/kg FW) was also measured. A comparison of the years 2021 and 2020 showed that the content was higher in 2020 than in 2021 (Appendix A).

The effect of the year could be evaluated only with the genotypes in which the hips could be harvested in both years. Comparing the phenolic compounds in the flesh with skin in hybrid *R*. × *R*. *glauca* and *R*. *corymbifera*, a significant interaction between genotype and both years was present, except in the case of flavones. The effect of the year was significant here. Both genotypes accumulated higher amounts of flavones in the first experimental year, 2020 (Table 3).

In the case of HBA, flavanols and flavones, the interaction between genotype and experimental year regarding values in seeds was significant. The HCA values, values of gallotannins and ellagitannins in seeds of hips were genotype-dependent. Seeds of rose hips in the *R.* × *R. glauca* hybrid contained significantly higher HCA and gallotannin values than the seeds of *R. corymbifera*, whereas the levels of ellagitannins were significantly higher in seeds of *R*. *corymbifera.* For flavonols, the year significantly affected the results since the accumulation of substances from this group was significantly higher in rosehip seeds in 2021 (Table 4).

Of anthocyanins, cyanidin-3-glucoside was determined in the rose hips. The highest content of cyanidin-3-glucoside was analyzed in *R. gallica* (28.78 mg/kg FW) in 2021 and the lowest in *R. subcanina* in the same year (1.12 mg/kg FW) (Figure 7 and Table 5).

Comparing the anthocyanin values in hips of *R.* × *R. glauca* and *R. corymbifera* in the two years, the interaction between year and both genotypes was significant. In 2020, the content of cyanidin-3-glucoside was quite high in *R.* × *R. glauca* (13.26 mg/kg FW), whereas the content in hips of *R. corymbifera* reached 5.75 mg/kg FW, and it was also almost the same in hips of the same species in 2021 (Table 5).

## 4. Discussion

During the present study, we identified 104 different phenolic compounds in rose hips. Predominant compounds were flavanols (33), followed by flavonols (23) and other compound groups with a lower number of members. The highest total phenolic content was determined in the flesh with skin in 2021 in *R. gallica* (15,767.21 mg/kg FW). The lowest content, also in 2021, was in the flesh with skin in hips of the *R. corymbifera*. The total content of phenolic compounds was overall lower in seeds than in the flesh with skin (1263.08 mg/kg FW–3247.89 mg/kg FW). Overall, the genotype *R. subcanina* was shown to be a proper control genotype in our study, similar to *R. canina* in various other studies. The total phenolic amount in different hips’ parts was the lowest in this control genotype.

Comparing the profile picture of a substance or group in the specific organ of a plant in one study with pictures in other studies is very difficult. The main problem is that these profile pictures are strongly dependent on various parameters, such as climate and soil conditions, age of the plant, etc. Additionally, the data concerning the amount of a substance have been considered in different units. This unit is very often the dry weight (DW) of the material to avoid problems with the amount of water in the material. We used fresh weight (FW) as the unit in our study because the analyzed material was rose hips, whose consumption is very often fresh and fresh weight is, therefore, a more appropriate form for consideration.

All these obstacles must be considered when comparing our results with those of other studies. In general, similar results to ours have been obtained in numerous studies with respect to the number of total phenolics. Najda and Buczkowska [27] measured the content of total phenolics in *R. californica*, *R. villosa, R. rugosa, R. spinosissima* and *R.* × *damascena*. The lowest total phenolic content was measured in *R.* × *damascena*, 109.67 mg/100 g FW, and the highest in *R. rugosa*, 215.14 mg/100 g FW. In our study, only *R. gallica* stood out from the mentioned range, with higher values of total phenolics in the flesh with skin. Ercisli [28] reported the total content of phenolic compounds in the fruits of *Rosa canina*, *Rosa dumalis* subsp. *boissieri*, *Rosa dumalis* subsp. *antalyensis*, *Rosa villosa*, *Rosa pulverulenta* and *Rosa pisiformis*. The highest total phenolic content was observed in *R. canina*, 96 mg GAE/g DW, and the lowest in *R. villosa*, 73 mg GAE/g DW. The phenolic composition of *R. canina* and *R. arvensis* is reported by Nadpal et al. [29], who found that the total content of phenolic compounds ranged from 6.63 mg GAE/g dry weight to 96.2 mg GAE/g dry weight. Shameh et al. [30] report lower values than the previously mentioned authors. Their values were between 3.31 mg GAE/g DW and 8.17 mg GAE/g DW. The chemical composition of phenolics in rosehip samples of *Rosa canina*, *Rosa corymbifera*, *Rosa micrantha* and *Rosa sempervirens* was evaluated by Fascella et al. [12], who reported that the total amount of phenolic compounds ranged from 40.58 mg GAE/g dry weight to 67.85 mg GAE/g dry weight. Ersoy et al. [13] reported that in rosehip samples of 25 Rosa genotypes, total phenolics ranged from 20.12 mg GAE/g DW to 32.20 mg GAE/g DW.

All samples in our study except *R. corymbifera* (flesh with skin and seeds) and *R. subcanina* (flesh with skin) had the highest content of procyanidin dimer 1. In the flesh with skin, the values varied between 1.18 g/kg FW and 1.85 g/kg FW, and in seeds, between 0.15 g/kg FW and 0.7 g/kg FW. In *R. corymbifera*, the predominant compound was procyanidin trimer 1 (2.99 g/kg FW) in the flesh with skin and catechin (0.2 g/kg FW) in the seeds. Cunja et al. [31] studied the phenolic compound content in *R. canina* harvested at six different times, every two weeks from early September to early November and again in early December when they were affected by cold weather. Flavanols were also the most prevalent among them, the content ranging between 3060.8 µg/g DW and 4217.3 µg/g DW. Catechin contents were reported to be between 79 µg/g DW and 124.6 µg/g DW. Guimarães et al. [32] reported that the content of proanthocyanidin (PA) dimers in the fruits of *R. canina* were 3.27 g/100 g and *in R. micrantha* 4.93 g/100 g. In both analyzed samples, procyanidins were the dominant phenolics. Ghendov-Mosanu et al. [33] found that the content of procyanidin B1 in *R. canina* was 29.1 g/100 g. Elmastas et al. [34] studied the phenolic composition in the fruits of *R. dumalis, R. canina* and *R. villosa*. On all harvest dates, catechin was the predominant flavanol, its values ranging from 269.08 mg/kg to 344.58 mg/kg in *R. dumalis*, while in *R. canina* these values ranged from 225 mg/kg to 437 mg/kg and in *R. villosa* from 343.58 mg/kg to 416.50 mg/kg. In our study, the values in the flesh with skin ranged between 116.03 mg/kg FW (*R. corymbifera*) and 612.31 mg/kg FW (*R.* × *R. glauca*) and were closest to the range of values determined by Elmastas et al. [34] in *R. canina*. Goztepe et al. [35] determined the catechin content of rose hips from Turkey to be 347 mg/100 g FW. Ghendov-Mosanu et al. [33] also found that catechin was one of the more abundant compounds in *R. canina* hips. Its value was, on average, 4.6 mg/100 g. Epicatechin was present in even higher concentrations (5.7 mg/100 g). In our samples, however, epicatechin was present in the highest amounts in the flesh with the skin of the hips of *R. gallica* (329.67 mg/kg). Demir et al. [36] reported a high content of gallic acid (7.70 µg/g DW), which was present in our study in hybrid *R.* × *R. glauca* (flesh with skin and seeds) in seeds of *R. corymbifera* and *R. subcanina* and in the flesh with skin in hips of *R. gallica*.

Guimarães et al. [32] studied *R. canina* and *R. micrantha*. The total content of phenolic acids and flavones/ols was 5.50 mg/100 g in *R. canina* and 11.16 mg/100 g in *R. micrantha*. Among them, taxifolin pentoside was predominant in both roses, 1.18 mg/100 g in *R. canina* and 2.68 mg/100 g in *R. micrantha*. In our samples, taxifolin pentoside 1 was present only in the flesh with the skin of hips of *R. corymbifera* (in both analyzed years) and in the flesh with the skin of hips of *R. subcanina,* but we did not detect taxifolin pentoside 1 in the seeds in our study.

In seeds, the dominant compound among HBAs in *R. corymbifera* was ellagic acid pentoside 1 (159.30 mg/kg FW, in 2020), in the hybrid with *R. glauca* galloyl quinic acid (in the flesh with skin), was predominant (178.90 mg/kg FW) and gallic acid in seeds (103.01 mg/kg FW). In *R. subcanina*, taxifolin pentoside 3 (6.80 mg/kg FW) and gallic acid (67.30 mg/kg FW) predominated in the flesh with skin. In *R. gallica*, the situation was just the opposite. Gallic acid (62.30 mg/kg FW) predominated in the flesh with skin, while ellagic acid pentoside 1 (2.95 mg/kg FW) predominated in the seeds. Cunja et al. [31] reported the content of apigenin derivatives 1 and 2. They reported that the total content of the mentioned flavones was between 4.1 and 8.5 µg/g DW. We also determined these two derivatives, which were present in the flesh with skin. Their values in the flesh with skin ranged between 0.12 mg/kg FW (*R. gallica*) and 9.04 mg/kg FW (*R. subcanina*). In seeds of the hybrid with *R. glauca*, the value of apigenin derivative 1 was 5.04 mg/kg FW in 2021 and 1.16 mg/kg FW in 2020.

Among HCAs, sinapic acid hexoside 2 predominated in the flesh with skin of the hybrid with *R. glauca* in both years and sinapic acid hexoside 1 in the seeds in 2021. In the flesh of *R. corymbifera* in 2020, sinapic acid hexoside 1 and 5-*p*-coumaroylquinic acid 2 were the predominant substances in seeds. In *R. subcanina*, sinapic acid hexoside 1 was predominant in the flesh with skin and 5-caffeoylquinic acid 2 in the seeds. In the last analyzed rose, *R. gallica*, 5-*p*-coumaroylquinic acid 1 predominated in the flesh with skin, and *p*-coumaric acid hexoside 1 in the seeds. Cunja et al. [31] also listed methyl gallate hexoside (67.7–97.6 µg/g DW), sinapic acid hexoside (31.8–44.0 µg/g DW) and ellagic acid penoside 1 as the main components of phenolic acids and their derivatives.

Among gallotannins, methyl gallate hexoside (7.01 mg/kg FW) was present in the largest amount in the flesh with the skin of *R. corymbifera* in 2021, while digalloyl hexoside 2 was present in the lowest amount, also in *R. corymbifera*, but in seeds in the same year. Guimarães et al. [32] reported that the total content of flavan-3-ols in *R. canina* was 19.90 mg/100 g and in *R. micrantha* 32.62 mg/100 g. The predominant compound in *R. canina* was catechin (3.59 mg/100 g), while in *R. micrantha* it was PA dimer (4.93 mg/100 g). Cunja et al. [31] reported that in addition to proanthocyanidins, catechin and catechin hexoside were also quantified in rose hips. However, their content changed drastically during the ripening process. The highest content was determined on the third sampling date (124.6 µg/g DW catechin and 315.9 µg/g DW catechin hexoside), then decreased and was the lowest at the flank freezing stage (79.0 µg/g DW catechin and 178.3 µg/g DW catechin hexoside). However, due to the high proportion of proanthocyanidins, only minor differences in total flavanol concentrations were observed in rose hips during fruit ripening. In our case, the content of catechin in the flesh with skin ranged from 116.03 mg/kg FW (*R. corymbifera*) to 612.31 mg/kg FW (hybrid with *R. glauca*). In seeds, however, the value varied between 6.29 mg/kg FW (*R. gallica*) and 199.83 mg/kg FW (*R. corymbifera*). Catechin hexoside, cited by Cunja et al. [31], was only present in the seeds of the hybrid with *R. glauca* (120.04–179.04 mg/kg FW), in the seeds of *R. subcanina* (0.18 mg/kg) and in the seeds of *R. gallica* (124.03 mg/kg) in our study. The highest PA dimer diglycoside content, 522.66 mg/kg FW, was measured in 2020 in *R. corymbifera*. In our research, the flavonol with the highest concentration was quercetin-3-rutinoside (371.06 mg/kg FW), which was present in the hybrid with *R. glauca*. Guimarães et al. [32] reported that the content of quercetin-3-glucoside in *R. canina* was 0.47 mg/100 g, and in *R. micrantha* 0.32 mg/100 g. Among flavanols, we identified apigenin derivatives 1 and 2, as did Cunja et al. [19]. They were present in the seeds only in the hybrid with *R. glauca* (0.001 mg/kg FW–0.005 mg/kg) and in the flesh of all analyzed rosehips, ranging between 0.001 mg/kg FW and 0.008 mg/kg FW. Of dihydrochalcones, only phloridzin was identified, which was only present in the flesh with the skin of the analyzed rosehips but not in the seeds. Its content was strongly dependent on the genotype. In the hips of *R.* × *R. glauca*, it was present in very small amounts, 0.08 µg/kg FW, while the hips of *R. gallica* contained much more, 1.1 mg/kg FW. Derivatives of quercetin are mainly present in rose hips: quercitrin (quercetin-3-*O*-rhamnoside), isoquercitin (quercetin-3-*O*-glucoside) and hyperoside (quercetin-3-*O*-galactoside) [37,38]. It is known that there are species differences in the content and composition of flavonol compounds among rose hips. However, precisely because of such great intra- and inter-specific variability of the section Caninae, this area is still relatively unexplored [39,40,41,42]. The phloridzin content in the hips of roses from the study of Cunja et al. [31] ranged between 19.4 µg/g DW and 47.7 µg/g DW. Elmastaş et al. [34] studied *Rosa dumalis, R. canina* and *R. villosa*, the hips of which were harvested at different times based on color changes during the ripening period. The main compounds determined were phenolic acid derivatives (caffeic, ferulic, gallic and *p*-coumaric acid) and flavonoids (kaempferol, eriocitrin, catechin, apigenin-7-*O*-glucoside, quercetin, apigenin and rutin). The content of phenolic acids of *Rosa* species increased non-linearly with respect to the time of harvesting. The maximum amount of catechin was found at the fifth harvest time point, ranging from 323 to 472 mg/kg. The highest content of caffeic acid was found in *R. dumalis,* ranging from 24 to 77 mg/kg. During the study, we determined that 3-feruloyquinic acid was present only in the flesh with skin (1–8 µg/kg FW) and in seeds (2–15 µg/kg FW) in the hybrid with *R. glauca*. Nadpal et al. [29] reported that ferulic acid was absent in a fresh methanolic extract of *R. canina* but was present in *R. arvensis* (10.20 µg/g). Fascella et al. [12] determined the phenolic composition of *R. canina, R. corymbifera, R. micrantha* and *R. sempervirens* hips. They found that the highest content of quercetin-3-glucoside and quercetin-3-glucuronide was in the hips of *R. canina*, followed immediately by *R. corymbifera*. In our case, the content of quercetin-3-glucoside was highest in the flesh with skin in *R. subcanina* (40 µg/kg FW) in 2021. In the seeds, the content of this compound was highest (3 µg/kg FW) in *R. gallica* in 2021. The content of quercetin-3-glucuronide was highest in the flesh with skin and in the seeds of *R. corymbifera* in our study. In both cases, its value was 10 µg/kg FW. Nadpal et al. [29] determined that the content of quercetin-3-glucoside in an ethanolic extract of *R. arvensis* was 208 µg/g, while its content in *R. canina* was 9.40 µg/g. Their finding that *p*-coumaric acid was present only in *R. canina* (1.53 µg/g) was also interesting, while it was not detected in *R. arvensis*. In our study, *p*-coumaric acid was present in the hips of all genotypes.

The content of cyanidin-3-glucoside was highest in *R. gallica* in 2021 (28.78 mg/kg FW). A surprisingly low value was measured for *R. subcanina* (in 2021), only 1.13 mg/kg FW. Comparing the hybrid plants with *R. glauca* in the years 2021 and 2020, it was observed that the content was much higher in 2020 (13.26 mg/kg FW) than in 2021 (5.18 mg/kg FW), whereas the anthocyanin concentration in hips of *R. corymbifera* was greater in 2021 (7.01 mg/kg FW in 2021 compared to 5.75 mg/kg FW in 2020). This is clear evidence that microclimatic conditions are very important for anthocyanin synthesis. Guimarães et al. [32] found that *R. canina* contained 0.68 µg/100 g DW cyanidin-3-*O*-glucoside and *R. micrantha* 1.19 µg/100 g DW. Cunja et al. [19] determined the content of cyanidin glucoside in *R. canina*, *R. glauca*, *R. sempervirens* and *R. rubiginosa*. The anthocyanin content in the hips of *R. glauca* was 26.2 µg/g FW. The content of anthocyanin in the hips of *R. gallica* in our study was higher than its content in the hips of *R. pendulina* in the study of Kunc et al. [24]. However, the content of anthocyanin in the hips of cross *R. pendulina* × *spinosissima* was even much higher (40 mg/kg FW) in this study. This is evidence that first progenies from the cross of two genotypes can have enhanced accumulation of some metabolites. Fascella et al. [12] reported that *R. canina* and *R. micrantha* hips had the highest total anthocyanin contents (2.94 and 3.86 mg CGE/100 g DW, respectively). Based on our radiation measurements, which did not differ significantly between locations, we cannot explain such a low number of anthocyanins in *R. subcanina* as being due to low radiation since the bushes of *R. subcanina* and *R. gallica* were at the same location, a few meters apart, and under the same average radiation. As a result, all weather conditions were also the same. Radiation was also not significantly different between locations, so it could not have had such a strong influence on the differences in anthocyanin content between *R. gallica* and the other rosehips studied. It can, therefore, be claimed that, in our case, the influence of the environment and various ecological influences on the plants was negligible. Because these are plants growing in nature, their age cannot be determined, so we assume that the higher anthocyanin concentration is also likely due in part to the different ages of the plants. We assume that *R. subcanina* was a younger, less developed plant and that it consequently had lower anthocyanin contents and a less developed root system than surrounding plants, depriving it of nutrients. As a result, it accumulated anthocyanins in lower amounts. However, in this discussion, we have excluded the influence of diseases and pests, which can also stimulate the formation of anthocyanins [43]. Consequently, also in our study, the effect of genotype is even more important than the age of a plant in relation to the mentioned differences in polyphenolic profiles, as has been frequently reported in previous studies [44,45].

Comparing the content of total phenolic compounds in *R.* × *R. glauca* and *R. corymbifera* between 2021 and 2020, it was found that the content in the flesh with skin decreased by about half in the second year of observation. In the seeds, the content increased by a factor of about one. In terms of the locations, it can be seen that the content in the flesh with skin was very similar between *R.* × *R. glauca*, which was in the Nanos region, and *R. corymbifera*, which was growing in Zagorje near Pivka. The seeds of *R.* × *R. glauca* had a slightly higher content of total phenolics. The lowest content of phenolics in the flesh with skin was found in *R. corymbifera* and the highest in *R. gallica*. The higher content of phenolic compounds in 2020, in Zagorje near Pivka and Nanos, is probably the result of the weather conditions in 2020, which were more favorable for the formation of phenolic compounds in the flesh of hips. In 2021, however, they were more favorable for the formation of phenolic compounds in the seeds. In 2020, both Zagorje near Pivka and Nanos experienced higher winter temperatures and longer-lasting higher summer temperatures than in 2021. During the ripening season in autumn, temperatures dropped faster in 2020 than in 2021. The greater formation and accumulation of phenolics, particularly anthocyanins, in 2020 was due to a sufficiently large difference between lower and higher temperatures in that year compared to 2021 [46]. In 2020, there was less precipitation in Zagorje near Pivka (but more in the summer months), while in Nanos, there was more precipitation at once in 2021 than in 2020, but it was much less well distributed among the months compared to 2020 (the highest amount fell in May). More precipitation fell during fruit ripening in Zagorje near Pivka in 2020 than during the ripening period in 2021, and almost twice as much rain fell on Nanos during the ripening period in 2020 as in 2021.

Based on the obtained values of the content of phenolic compounds in the studied rose hips, we confirm that they are a good source of phenolic compounds that have a beneficial effect on our health. Willich et al. [10] supported the research by studying the effect of *R*. *canina* powder on rheumatoid arthritis. The study involved patients who were prescribed rosehip capsules and another group who received a placebo. They found that the condition of the patients who received the rosehip capsules improved, while the condition of the other group of patients worsened. Marmol et al. [8] described that *Rosa* spp. extracts have a similar phytochemical profile to blueberries, which are characterized by high antioxidant content, so they also play an important role in the prevention of osteoporosis. In addition, they found that *R*. *canina* extract significantly increases the proliferation of pancreatic β-cells, which lowers blood glucose levels [8]. Cancer development is strongly associated with intracellular levels of reactive oxygen species. The role of rosehip in this context is to reduce cell viability after incubation of these cancer cell lines with the whole rosehip extract or with the purified fractions of phenolic compounds [8]. Tamura et al. [47] investigated the activity of *R*. *rugosa* against the hepatitis C virus. They found that phytochemicals from rosehip inhibited the ability of the hepatitis C virus to invade hepatocytes. Baie et al. [48] reported the antimicrobial effect of *R*. *indica* on urease. They found that it prevented the formation of crystals and stones, indirectly preventing nephrolithiasis and other kidney damage.

## 5. Conclusions

The results of the study provide information on the content and accumulation of phenolic compounds in native-grown rosehips *R.* × *R. glauca*, *R. corymbifera, R. gallica* and *R. subcanina*, which was elected as the native control genotype, and all grow in southwestern Slovenia. These major rose genotypes are still largely uninvestigated. We found that *R. gallica* was the richest in various phenolics, followed by *R. subcanina, R.* × *R. glauca* and *R. corymbifera*. The assumption that the total content of phenolic compounds is higher in the flesh with skin than in the seeds was confirmed. The highest number of different identified phenolic compounds was found in the rose hips of hybrid *R.* × *R. glauca*, both in the skin with flesh and in the seeds. The lowest number was found in rose hips of *R. gallica*, which contained a high content of anthocyanin, cyanidin-3-glucoside, in addition to the highest total phenolic content compared to other genotypes. The year 2020 was more favorable for the formation of phenolic compounds in the pulp with rosehip skin, and the year 2021 was for the accumulation of phenols in the seeds. Higher winter temperatures, longer-lasting higher summer temperatures, a faster temperature drop during ripening, a greater difference between low and high temperatures, and higher rainfall in 2020 contributed to the higher formation of phenolic compounds in 2020.

In conclusion, the rose hips of the different rose genotypes studied are a rich source of various phenolic compounds and could potentially be used as functional foods. The present study encourages further analysis of bioactive compounds, which, in addition to phenolic compounds, are among the key compounds that could contribute to rose hips being an important component of a successful and healthy diet.

## Figures and Tables

**Figure 1 foods-12-01952-f001:**
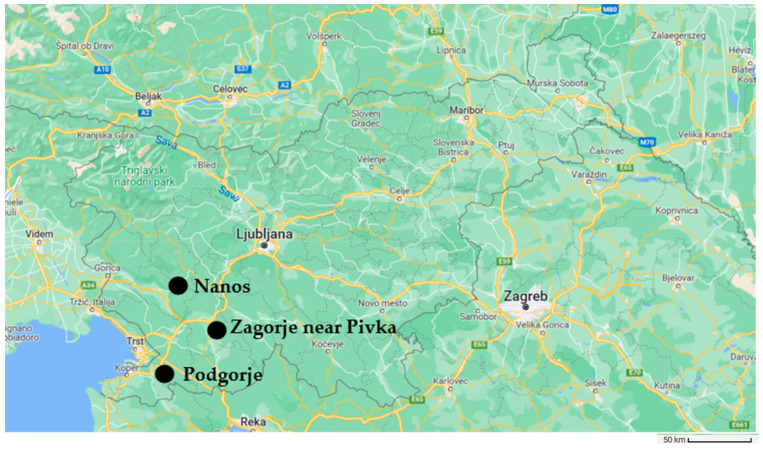
Display of the locations (Zagorje near Pivka, Nanos and Podgorje) of the collected material.

**Figure 2 foods-12-01952-f002:**
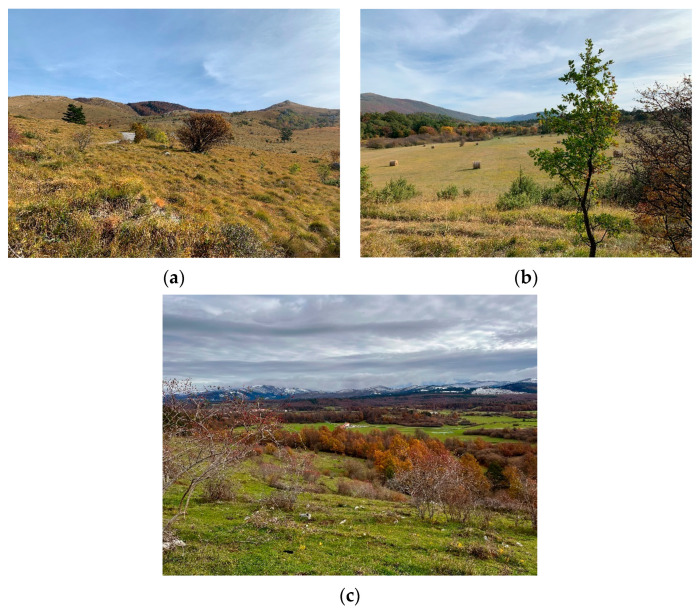
Relief variability of different rose deposits (Nanos (**a**), Podgorje (**b**) and Zagorje near Pivka (**c**)).

**Figure 3 foods-12-01952-f003:**
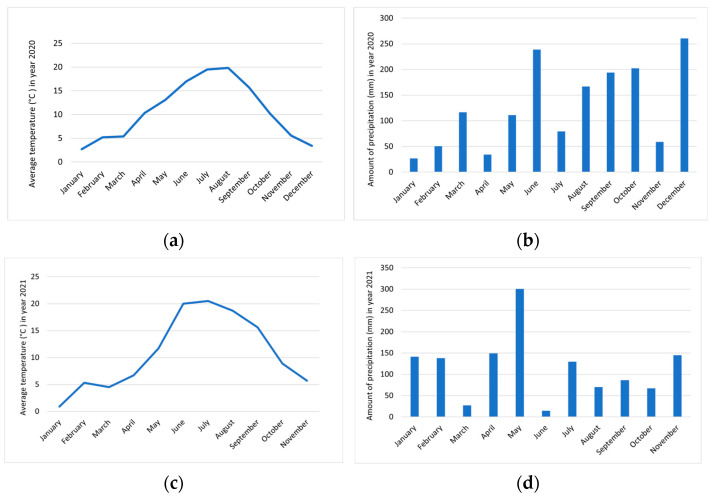
Weather conditions in the area Zagorje near Pivka during the duration of our study. (**a**) Average temperature (°C) in 2020, (**b**) amount of precipitation (mm) in 2020, (**c**) average temperature (°C) in 2021 and (**d**) amount of precipitation (mm) in 2021 [23].

**Figure 4 foods-12-01952-f004:**
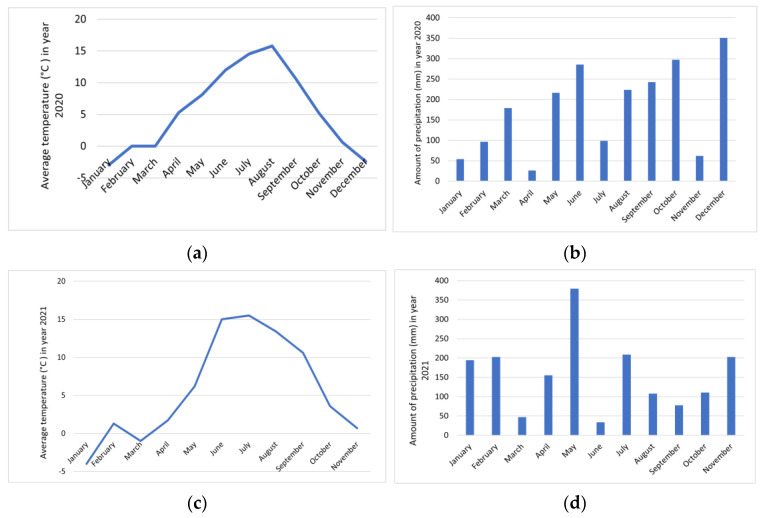
Weather conditions in the area of Nanos during the duration of our study. (**a**) Average temperature (°C) in 2020, (**b**) amount of precipitation (mm) in 2020, (**c**) average temperature (°C) in 2021 and (**d**) amount of precipitation (mm) in 2021 [23].

**Figure 5 foods-12-01952-f005:**
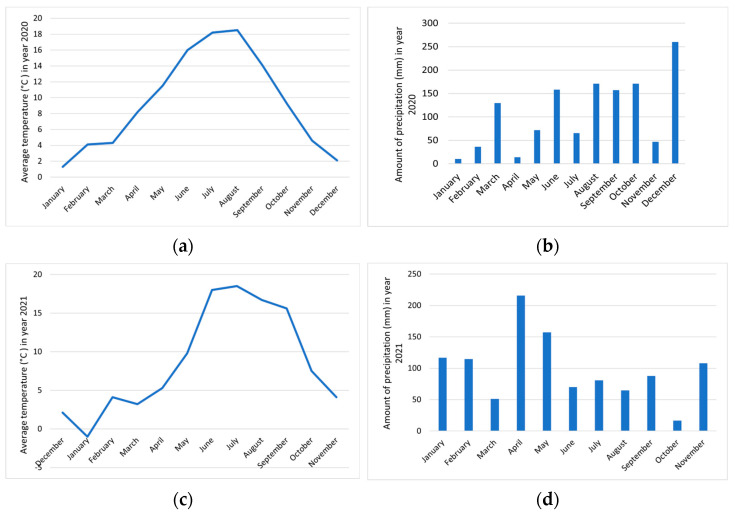
Weather conditions in the area of Podgorje during the duration of our study. (**a**) Average temperature (°C) in 2020, (**b**) amount of precipitation (mm) in 2020, (**c**) average temperature (°C) in 2021 and (**d**) amount of precipitation (mm) in year 2021 [23].

**Figure 6 foods-12-01952-f006:**
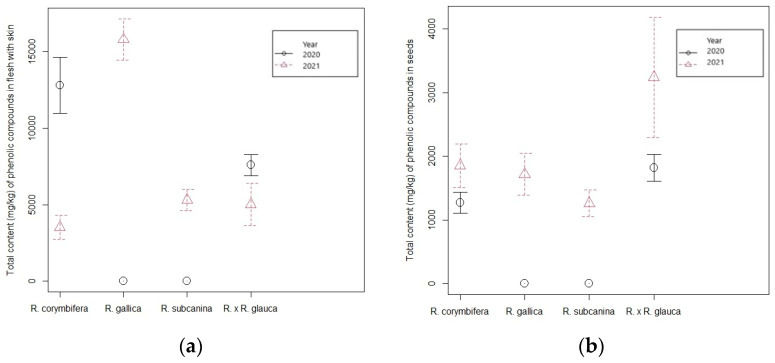
Total content of phenolic compounds in (**a**) flesh with skin and in (**b**) seeds of *R*. × *R*. *glauca* and *R*. *corymbifera* over a period of two years (2020–2021) and in 2021 for *R. gallica* and *R. subcanina* ± standard error (mg/kg FW).

**Figure 7 foods-12-01952-f007:**
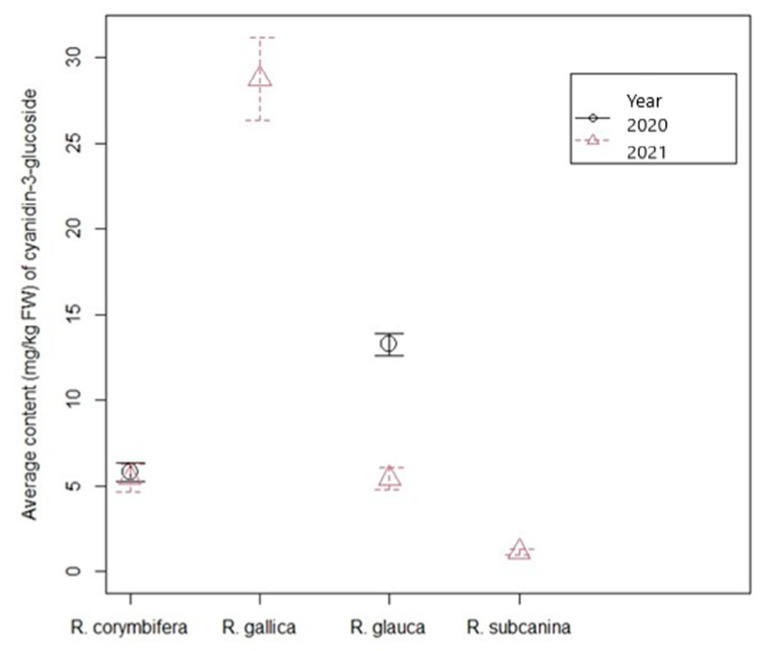
Average contents ± standard error (mg/kg FW) of cyanidin-3-glucoside in rosehips of *R.* × *R. glauca* and *R. corymbifera* over a period of two years (2020–2021) and *R. gallica* and *R. subcanina* in 2021.

**Table 1 foods-12-01952-t001:** Average photosynthetic active radiation (PAR) (W/m^2^) and average intensity of light (W/m^2^) in the range 400–500 nm, 500–600 nm, 600–700 nm in Podgorje, Zagorje near Pivka and Nanos, respectively.

	Podgorje	Zagorje Near Pivka	Nanos
Photosynthetic active radiation (PAR)	1481	1595.5	1655.6
400–500 nm	423.9	444.4	544.7
500–600 nm	535.1	559.4	556.7
600–700 nm	594.8	603.2	595.1

**Table 2 foods-12-01952-t002:** Total content of phenolic compounds in flesh with skin and in seeds of *R.* × *R. glauca* and *R. corymbifera* over a period of two years (2020–2021) ± standard error (mg/kg FW) and in 2021 for *R. gallica* and *R. subcanina.* Different lowercase letters indicate statistically significant differences between species each year (separately for flesh with skin and seeds).

Year	*R.* × *R. glauca*		*R. corymbifera*		*R. subcanina*		*R. gallica*	
	Flesh and Skin	Seeds	Flesh and Skin	Seeds	Flesh and Skin	Seeds	Flesh and Skin	Seeds
2021	5010.01 ± 4102.08 b	3247.89 ± 2167.13 c	3501.38 ± 1909.57 a	1851.83 ± 995.51 b	5305.45 ± 1892.66 b	1263.08 ± 651.09 a	15,767.21 ± 3628.34 c	1711.60 ± 821.49 b
2020	7650.93 ± 3042.83 a	1821.24 ± 920.47 b	12,832.94 ± 6505.10 b	1285.14 ± 633.16 a	-	-	-	-

Note: (-) *R. subcanina* and *R. gallica* did not produce hips in 2020.

**Table 3 foods-12-01952-t003:** Comparison of the main phenolic groups content (mg/kg FW ± SE) between the two analyzed years (2021 and 2020) in flesh with skin of *R.* × *R. glauca* and *R. corymbifera*. Different lowercase letters indicate statistical differences between genotypes and years, and different capital letters indicate statistically significant differences between years (Tukey test at ‘***’ = 0.001; ‘**’ = 0.01; ‘*’ = 0.05; NS ‘no statistical difference’).

					Compounds				
Genotype	Year	HBA	HCA	Gallotannins	Ellagitannins	Flavanols	Flavonols	Flavones	Dihydrochalcone
*R.* × *R. glauca*	2021	70.30 ± 20.01 a	67.31 ± 29.36 a	377.36 ± 221.21 a	727.54 ± 241.77 b	3695.83 ± 3541.84 a	57.48 ± 45.09 a	8.23 ± 1.06 A	0.78 ± 0.04 ab
	2020	222.05 ± 102.77 b	103.33 ± 48.9 a	681.30 ± 352.46 a	638.73 ± 248.69 ab	5848.32 ± 2201.74 a	76.87 ± 46.80 a	8.43 ± 3.17 B	58.64 ± 36.02 c
*R. corymbifera*	2021	43.65 ± 13.57 a	171.79 ± 94.72 a	261.61 ± 129.8 a	366.51 ± 136.78 a	2579.61 ± 1487.31 a	65.48 ± 40.29 a	5.21 ± 3.72 A	0.51 ± 0.44 a
	2020	62.50 ± 30.56 a	769.16 ± 413.18 b	1396.09 ± 515.09 b	911.55 ± 468.95 b	9358.54 ± 4929.68 b	273.28 ± 126.58 b	9.04 ± 6.32 B	47.02 ± 13.11 b
	Genotype	-	-	-	-	-	-	NS	-
	Year	-	-	-	-	-	-	**	-
	Interaction	***	***	***	*	**	**	NS	***

**Table 4 foods-12-01952-t004:** Comparison of the main phenolic groups content (mg/kg FW ± SE) between the two analyzed years (2021 and 2020) in seeds of *R.* × *R. glauca* and *R. corymbifera*. Different lowercase letters indicate statistical differences between genotypes in each year, and different capital letters indicate statistically significant differences between genotypes (A, B) or years (X, Y) (Tukey test at ‘***’ = 0.001; ‘**’ = 0.01; ‘*’ = 0.05; NS ‘no statistical difference’).

Compound								
Genotype	Year	HBA	HCA	Gallotannins	Ellagitannins	Flavanols	Flavonols	Flavones
*R.* × *R. glauca*	2021	113.91 ± 67.39 ab	136.00 ± 56.67 B	270.60 ± 121.57 B	213.50 ± 152.53 A	2145.76 ± 1490.84 b	363.08 ± 276.07 Y	5.04 ± 2.06 b
	2020	103.48 ± 73.23 a	79.88 ± 38.75 B	270.02 ± 162.24 B	151.34 ± 72.05 A	1119.43 ± 516.31 a	95.93 ± 57.86 X	1.16 ± 0.03 a
*R. corymbifera*	2021	39.70 ± 26.63 a	57.61 ± 26.33 A	141.40 ± 75.77 A	112.61 ± 34.42 B	973.65 ± 572.06 a	526.86 ± 260.30 Y	0 ± 0 a
	2020	173.56 ± 67.48 b	56.41 ± 18.46 A	45.70 ± 29.83 A	126.43 ± 69.38 B	795.11 ± 392.95 a	89.93 ± 55.06 X	0 ± 0 a
	Genotype	-	*	***	***	-	NS	-
	Year	-	NS	NS	NS	-	***	-
	Interaction	**	NS	NS	NS	*	NS	*

**Table 5 foods-12-01952-t005:** Average contents ± standard error (mg/kg FW) of cyanidin-3-glucoside in rosehips of *R.* × *R. glauca* and *R. corymbifera* over a period of two years (2020–2021) and *R. gallica* and *R. subcanina* in 2021. Different lowercase letters indicate statistical differences between genotypes in 2021. Different capital letters indicate statistically significant differences between genotypes *R.* × *R. glauca* and *R. corymbifera* in the experimental years.

	*R.* × *R. glauca*	*R. corymbifera*	*R. subcanina*	*R. gallica*
2021	5.18 ± 1.7 bA	7.01 ± 2.94 bA	1.13 ± 0.45 a	28.78 ± 5.91 c
2020	13.26 ± 2.28 bB	5.75 ± 1.63 aA	-	-
Genotype	***	-		
Year	-	-		
Interaction	-	***		

Note: (-) Compound was not detected; Signif. codes: 0 ‘***’.

## Data Availability

All data are presented within the article.

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
