# Peer review of "Phenolic Compounds of Rose Hips of Some Rosa Species and Their Hybrids Native Grown in the South-West of Slovenia during a Two-Year Period (2020–2021)"

_foods, 2023, doi:10.3390/foods12101952_

Round 1

Reviewer 1 Report

The manuscript ID Foods: 2349152 submitted by Nina Kunc et al. describes the composition of major phenolics in the hips of different main rose species (R. corymbifera, R. gallica, R. subcanina and an unknown cross with R. glauca) which are distributed over the territory of Slovenia. The manuscript has valuble results related to HPLC - MS analysis of the phenolic compounds contained in the pulp and seed of native grown rose hips in southwestern of Slovenia. A comparison of phenolic composition measured separately  in the flesh with skin and seed of the rose hips between the two years of study and their connection with environmental conditions of growing locations represent a new approach with recently increased interest of researches.

The authors reported that the hips of the different rose genotypes are a rich source of various phenolic compounds, which could potentially be used as functional foods.

However, the manuscript needs some revisions as suggested in pdf version of the manuscript.

Reviewer 2 Report

The introduction should be improved by highlighting the importance of phenolic compounds in foods and in particular those derived from rose hips. In the same way in the discussion and with collusions, the results obtained in the research on the impact they will have on the foods that can be formulated or developed from the rose hips should be highlighted. On the other hand, the quality of Figure 6 and 7 should be improved.

Reviewer 3 Report

Introduction session is quite confusing. Most of information reported must be putted in discussion session. More emphasis must be placed in the novelty of the research paper.

How authors carried out quantification of all the compounds identified? Do the authors have all the reference materials? Did the authors construct calibration curves for all molecules? This information is completely absent in the experimental conditions section. 

Other missing information concerns the validation of the analytical method (precision, accuracy, linearity, LoD, LoQ) and of the extraction method. This information is necessary in order to publish a scientific article in which the qualitative and quantitative content is compared in two different years and in different parts of the plant.

 Moderate editing of English language

Round 2

Reviewer 2 Report

The publication is accepted with the modifications made.

Reviewer 3 Report

Authors submitted the manuscript revised according to all the reviewers comment. Some information are reported in the response to reviewer letter.

Introduction was organized according to reviewer suggestion. Moreover authors add all the information on figure of merits.

Now the article is suitable for publication in Foods journal.

Moderate editing of English language